# Deep Learning-Driven Abbreviated Shoulder MRI Protocols: Diagnostic Accuracy in Clinical Practice

**DOI:** 10.3390/tomography11040048

**Published:** 2025-04-17

**Authors:** Giovanni Foti, Flavio Spoto, Thomas Mignolli, Alessandro Spezia, Luigi Romano, Guglielmo Manenti, Nicolò Cardobi, Paolo Avanzi

**Affiliations:** 1Department of Radiology, IRCCS Sacro Cuore Don Calabria Hospital, 37024 Negrar, Italythomas.mignolli@sacrocuore.it (T.M.); luigi.romano@sacrocuore.it (L.R.); 2Department of Radiology, Policlinico Universitario GB Rossi, 37134 Verona, Italy; alessandrospezia94@gmail.com (A.S.); nicolo.cardobi@gmail.com (N.C.); 3Department of Diagnostic Imaging and Interventional Radiology, University Hospital, Policlinico Tor Vergata, 00133 Rome, Italy; gu.manenti@gmail.com; 4Department of Orthopaedic Surgery, IRCCS Sacro Cuore Don Calabria Hospital, 37024 Negrar, Italy; paolo.avanzi@sacrocuore.it

**Keywords:** magnetic resonance imaging, deep learning, shoulder, diagnostic accuracy, accelerated protocols

## Abstract

Background: Deep learning (DL) reconstruction techniques have shown promise in reducing MRI acquisition times while maintaining image quality. However, the impact of different acceleration factors on diagnostic accuracy in shoulder MRI remains unexplored in clinical practice. Purpose: The purpose of this study was to evaluate the diagnostic accuracy of 2-fold and 4-fold DL-accelerated shoulder MRI protocols compared to standard protocols in clinical practice. Materials and Methods: In this prospective single-center study, 88 consecutive patients (49 males, 39 females; mean age, 51 years) underwent shoulder MRI examinations using standard, 2-fold (DL2), and 4-fold (DL4) accelerated protocols between June 2023 and January 2024. Four independent radiologists (experience range: 4–25 years) evaluated the presence of bone marrow edema (BME), rotator cuff tears, and labral lesions. The sensitivity, specificity, and interobserver agreement were calculated. Diagnostic confidence was assessed using a 4-point scale. The impact of reader experience was analyzed by stratifying the radiologists into ≤10 and >10 years of experience. Results: Both accelerated protocols demonstrated high diagnostic accuracy. For BME detection, DL2 and DL4 achieved 100% sensitivity and specificity. In rotator cuff evaluation, DL2 showed a sensitivity of 98–100% and specificity of 99–100%, while DL4 maintained a sensitivity of 95–98% and specificity of 99–100%. Labral tear detection showed perfect sensitivity (100%) with DL2 and slightly lower sensitivity (89–100%) with DL4. Interobserver agreement was excellent across the protocols (Kendall’s W = 0.92–0.98). Reader experience did not significantly impact diagnostic performance. The area under the ROC curve was 0.94 for DL2 and 0.90 for DL4 (*p* = 0.32). Clinical Implications: The implementation of DL-accelerated protocols, particularly DL2, could improve workflow efficiency by reducing acquisition times by 50% while maintaining diagnostic reliability. This could increase patient throughput and accessibility to MRI examinations without compromising diagnostic quality. Conclusions: DL-accelerated shoulder MRI protocols demonstrate high diagnostic accuracy, with DL2 showing performance nearly identical to that of the standard protocol. While DL4 maintains acceptable diagnostic accuracy, it shows a slight sensitivity reduction for subtle pathologies, particularly among less experienced readers. The DL2 protocol represents an optimal balance between acquisition time reduction and diagnostic confidence.

## 1. Introduction

Magnetic resonance imaging (MRI) of the shoulder is essential for evaluating shoulder joint pathologies [1,2,3,4,5]. The technology’s excellent soft tissue contrast and multiplanar acquisition capabilities enable optimal assessments of muscles, tendons, hyaline and fibrous cartilage, joint capsules, fat, bursae, and bone marrow [1,2,3,4,5]. Unlike ultrasound, which is effective primarily for assessing rotator cuff injuries, MRI offers a comprehensive evaluation of the bone marrow, cartilage, and glenoid labrum, establishing it as the most reliable imaging modality [3,4,5]. Common indications for shoulder MRI include suspected rotator cuff tears, shoulder instability, osteonecrosis, neoplasms, and infections. Additionally, MRI is proficient in diagnosing adhesive capsulitis and impingement syndromes [5,6,7].

Despite its advantages, shoulder MRI faces challenges, notably the high costs associated with prolonged acquisition times that elevate the risk of motion artifacts. This concern is particularly relevant when images with high spatial and contrast resolution are required to detect subtle tears in tendinous and ligamentous structures.

Recent advancements in artificial intelligence (AI), particularly deep learning (DL) algorithms, have been proposed to enhance image reconstruction from undersampled data, thereby reducing scanning times [8,9,10,11,12,13,14,15,16,17]. AI and DL techniques can be applied across various anatomical regions with different acceleration factors. These approaches differ from traditional acceleration techniques such as parallel imaging and compressed sensing in their ability to learn complex patterns from training data to reconstruct high-quality images from significantly undersampled acquisitions [9,10,11,12,13,14].

Deep learning reconstruction offers several advantages over conventional methods, including improved signal-to-noise ratios (SNRs) and reduced artifacts [8,15,16]. However, these benefits must be balanced against potential trade-offs in image quality, particularly at higher acceleration factors. While a modest acceleration factor may provide minimal time savings but preserve fine detail, implementing more aggressive acceleration—capable of reducing sequence durations by four to six times—could significantly enhance MRI workflows but potentially compromise the detection of subtle pathologies.

Different MR sequences may respond differently to DL reconstruction techniques. For instance, T1-weighted sequences, which typically have inherently high SNRs and relatively short acquisition times, may benefit less dramatically from DL reconstruction compared to T2-weighted or STIR sequences, which are often more signal-limited and time-consuming [8]. This differential benefit across sequence types is an important consideration when implementing accelerated protocols.

The recent literature has explored accelerated MRI protocols utilizing DL and AI for knee and shoulder examinations, revealing that DL can substantially reduce acquisition times while maintaining image quality and diagnostic confidence comparable to that of standard turbo spin-echo (TSE) MRI [17,18,19]. However, to date, there have been no studies comparing different acceleration levels from commercially available MRI scanners.

The objective of this study is to evaluate the diagnostic accuracy of 2-fold and 4-fold accelerated shoulder MRI protocols, using the standard protocol as the reference for diagnosis.

## 2. Materials and Methods

### 2.1. Study Population

Between June 2023 and January 2024, we considered 92 consecutive patients with clinically suspected rotator cuff tears, labral lesions, or bone edema during orthopedic or sports medicine visits for inclusion. The inclusion criteria were shoulder pain and no history of previous surgery. All patients underwent a standard MRI protocol (standard of care) as well as two accelerated protocols (ACS) on the same day.

The exclusion criteria included incomplete imaging data, examinations that were limited in their assessment due to significant motion artifacts, and previous surgery. Following the application of the exclusion criteria, four patients were removed from the analysis due to incomplete data acquisition (n = 2) and significant imaging artifacts (n = 2). The final study cohort comprised 88 patients, with a demographic distribution of 49 males and 39 females and a mean age of 51 years (range: 22–78 years).

### 2.2. MRI Standard Protocols

All patients were examined using a 3.0 Tesla MR scanner (uMR Omega, United Imaging Healthcare, Shanghai, China). The institutional review board validated the data collection for this prospective clinical study involving all adult patients (>18 years). All data were gathered as part of routine clinical care.

### 2.3. Technical Parameters

All MRI examinations were performed using a 16-channel dedicated shoulder coil. The standard protocol sequences included T1-weighted turbo spin-echo (TSE), T2-weighted TSE, and TIRM sequences in multiple planes. For T1-weighted TSE, the parameters were TR/TE/FA: 650.0 ms/18.0 ms/150°, slice thickness: 3 mm, acquisition planes: axial and coronal, acquisition time: 3 min 42 s per plane. For T2-weighted TSE, the parameters were TR/TE: 4300.0 ms/124.0 ms for coronal plane; TR/TE: 3500.0 ms/39.0 ms for axial plane, slice thickness: 3 mm, acquisition time: 4 min 18 s and 3 min 56 s, respectively. For TIRM sequences, the parameters were TR/TE: 4800.0 ms/46.0 ms, TI: 150 ms, slice thickness: 3 mm, acquisition time: 4 min 52 s.

The DL2 and DL4 protocols used identical base parameters as the standard protocol but incorporated the uAI deep learning reconstruction algorithm (United Imaging Healthcare) with acceleration factors of 2-fold and 4-fold, respectively. This resulted in the total acquisition times being reduced by 50% for the DL2 protocol and 75% for the DL4 protocol compared to the standard protocol. The specific acquisition times for DL2 were T1-weighted TSE: 1 min 51 s per plane, T2-weighted TSE: 2 min 9 s (coronal) and 1 min 58 s (axial), TIRM: 2 min 26 s. For DL4, the times were further reduced to T1-weighted TSE: 56 s per plane, T2-weighted TSE: 1 min 5 s (coronal) and 59 s (axial), TIRM: 1 min 13 s.

No modifications to the base sequence parameters were required for the accelerated protocols. The DL reconstruction was automatically applied during image acquisition using the manufacturer’s uAI algorithm, which employs a convolutional neural network trained on paired datasets of fully sampled and undersampled images [8,9,10,11,12,13]. Overall, the duration of each examination with all three protocols was shorter than the combined time of two standard protocols, and patients experienced no issues while remaining inside the gantry.

### 2.4. MRI Postprocessing

The accelerated protocols do not necessitate any additional post-acquisition processing since all acquisition parameters are comparable to those of the standard protocol. The MR system automatically employs the uAI deep learning reconstruction algorithm during image acquisition to reconstruct high-quality images from undersampled k-space data [14,15,16].

The uAI algorithm utilizes a multi-scale convolutional neural network architecture that was trained on paired datasets of fully sampled and undersampled acquisitions to learn optimal image reconstruction [8,15,16]. This approach differs from conventional parallel imaging and compressed sensing techniques by learning complex image features rather than relying on predefined mathematical constraints.

The DL reconstruction occurs in real time during the acquisition process, adding only minimal computational time (typically less than 10 s per sequence). There is no significant delay in obtaining and visualizing the accelerated images, ensuring that the workflow remains uninterrupted.

### 2.5. Image Analysis

Prior to the study, all four radiologists participated in a standardized training session that included an evaluation of 15 test cases (not included in the study cohort) to ensure consistent application of the evaluation criteria. This training session covered the specific scoring systems for each pathology and established agreement on borderline cases.

As the reference standard, all standard MRI images were evaluated in consensus by two experienced readers (L.R. and G.F.), who possess 25 and 15 years of experience, respectively, to determine the ground truth. Any disagreements between these two readers were resolved through discussion until consensus was reached. Subsequently, the 2-fold and 4-fold accelerated MRI (ACS) protocols were assessed on a dedicated offline workstation by four independent radiologists (A.S, E.O., L.R., and G.F., with 4, 10, 25, and 15 years of experience, respectively).

The evaluation was conducted blind and in random order. In the first step, image quality was assessed using a scale from 1 to 4, where 1 indicated poor quality, with the examination considered non-diagnostic, while 4 indicated optimal quality. Following this, the images were clinically evaluated for the diagnosis of structural abnormalities of the shoulder. Specifically, the readers assessed the presence of the following pathologies: bone edema, rotator cuff tears, and labral tears.

For each alteration, the results of the standard sequences and the accelerated sequences were deemed comparable only if there was a perfect anatomical match in the location of the alteration. All lesions were graded on a 4-point scale, where 1 indicated the absence of alterations and 4 indicated the presence of marked alterations.

The diagnosis of bone edema was based on the presence of increased water content, reflected by a signal increase in sequences with long repetition times (TRs) or signal decay in T1-weighted images. The scoring system was as follows: 1 for definitively no edema; 2 for presumably no edema; 3 for presumably the presence of edema; and 4 for definitively the presence of edema.

For tendon evaluation, the following scoring system was utilized: 1 for definitively no alterations; 2 for mild signal changes without tears; 3 for partial tears; and 4 for complete tears. Signal changes were noted in cases of tendon signal increases (in long-TR images), with or without thickening. Rotator cuff tears were identified by a loss of substance in one or more tendons of the cuff or the long head of the biceps. Each tendon was assessed individually, differentiating partial tears (articular or bursal) characterized by partial interruption of tendon fibers from complete ruptures, with or without tendon retraction. Given the non-arthrographic study protocol and the low incidence of labral abnormalities, the glenoid labrum was evaluated as a single anatomical unit without subdivision into its various anatomical sections. For labral tears, the following scoring system was adopted: 1 for definitively no alterations; 2 for mild signal changes without tears; 3 for partial tears; and 4 for complete tears.

### 2.6. Statistical Analysis

A statistical analysis was conducted using R software (version 4.3.0, R Foundation for Statistical Computing, Vienna, Austria). The sample size was determined assuming 90% power to detect a 10% difference in diagnostic accuracy between protocols (α = 0.05), requiring a minimum of 82 patients. A post hoc power analysis confirmed adequate statistical power (97%, β = 0.03) for detecting differences in diagnostic accuracy between protocols at the 0.05 significance level.

Standard protocol readings by two experienced radiologists (15 and 25 years of experience) served as the reference standard. For each reader and imaging parameter, we calculated the sensitivity, specificity, positive predictive value (PPV), negative predictive value (NPV), and accuracy with 95% confidence intervals (CIs).

Reader scores were dichotomized (scores 1–2 = negative, 3–4 = positive) for the statistical analysis. We constructed receiver operating characteristic (ROC) curves and calculated the area under the curve (AUC) to assess the overall diagnostic performance of each protocol. AUC values were computed and compared using DeLong’s test. McNemar’s test was used to assess paired differences in diagnostic accuracies between protocols.

Inter-reader agreement was evaluated using Kendall’s coefficient of concordance (W), with agreement strength classified as slight (0.00–0.20), fair (0.21–0.40), moderate (0.41–0.60), substantial (0.61–0.80), or almost perfect (0.81–1.00). We chose Kendall’s W over other agreement measures (such as kappa) because it can accommodate multiple raters simultaneously and provides a single consensus measure across all readers.

A secondary analysis stratified readers by experience level (≤10 years vs. >10 years) to assess whether reader experience moderated the effect of protocol on diagnostic performance. Diagnostic confidence scores were compared using the Kruskal–Wallis test with post hoc Dunn’s test, applying Benjamini–Hochberg correction for multiple comparisons to control the false discovery rate. Statistical significance was set at *p* < 0.05.

## 3. Results

### 3.1. Study Population Results

The initial study population consisted of 92 consecutive patients referred for shoulder MRI examination. Following the application of the exclusion criteria, four patients were removed from the analysis due to incomplete data acquisition (n = 2) and significant imaging artifacts (n = 2). The final study cohort comprised 88 patients, with a demographic distribution of 49 males and 39 females and a mean age of 51 years (range: 22–78 years).

### 3.2. Pathology Distribution in Reference Standard

An analysis of the reference-standard MRI sequences revealed bone marrow edema (BME) in 24 of the 88 cases (27.3%), while the remaining 64 patients showed no significant signal alterations. Examination of the rotator cuff demonstrated a total of 58 tears across the 352 evaluated tendons (16.5%). The distribution of these tears showed a predominance of supraspinatus (SST) involvement, with 38 tears identified, comprising 18 complete and 20 partial tears. The subscapularis (SSC) tendon exhibited 12 tears, equally divided between complete and partial involvement (6 each). Seven tears were identified in the infraspinatus (IST) tendon, including four complete and three partial tears. Additionally, a single complete tear of the teres minor was documented. Labral pathology was present in 9 of the 88 cases (10.2%) (Table 1).

### 3.3. Diagnostic Performance Analysis

#### 3.3.1. Bone Marrow Edema Detection

Both the DL2 and DL4 protocols achieved perfect diagnostic accuracy for BME detection, with all readers correctly identifying all cases (sensitivity 100%, 24/24; specificity 100%, 64/64). The 95% confidence intervals ranged from 85.8% to 100% for sensitivity and 94.4% to 100% for specificity.

#### 3.3.2. Rotator Cuff Evaluation

The DL2 protocol demonstrated marginally superior performance compared to DL4, though both maintained high diagnostic accuracy. DL2 showed sensitivity ranging from 98% to 100% (95% CI: 90.8–100%) and specificity from 99% to 100% (95% CI: 97.3–100%). The DL4 protocol maintained strong diagnostic capability with sensitivity between 95% and 98% (95% CI: 86.1–99.9%) and specificity from 99% to 100% (95% CI: 97.3–100%).

Individual tendon analyses revealed particularly strong performance in supraspinatus tear detection, with the DL2 protocol achieving 100% sensitivity and specificity across all readers. The DL4 protocol showed minimal variation in SST evaluation, with sensitivity ranging from 97% to 100% and specificity from 98% to 100%. Similar patterns were observed in subscapularis and infraspinatus evaluations, though the DL4 protocol showed slightly more variability in its sensitivity for IST tears (89–100%) (Table 2).

#### 3.3.3. Labral Tear Assessment

For labral tear detection, the DL2 protocol maintained perfect sensitivity (100%, 9/9) across all readers, with specificity ranging from 99% to 100%. The DL4 protocol showed slightly more variability in labral tear detection, with sensitivity ranging from 89% to 100%, while maintaining high specificity (99–100%) (Figure 1) (Table 3).

#### 3.3.4. Interobserver Agreement and Reader Experience Analysis

An interobserver agreement analysis demonstrated excellent consistency across both protocols. For the DL2 protocol, agreement coefficients reached almost perfect levels: 0.98 for BME evaluation (95% CI: 0.97–0.99), 0.95 for rotator cuff tears (95% CI: 0.93–0.97), and 0.96 for labral tears (95% CI: 0.94–0.98). The DL4 protocol maintained similarly strong agreement levels: 0.98 for BME (95% CI: 0.97–0.99), 0.92 for rotator cuff tears (95% CI: 0.89–0.95), and 0.93 for labral tears (95% CI: 0.90–0.96).

An analysis stratified by reader experience revealed no significant differences in diagnostic performance between readers with ≤10 years of experience (n = 2) and those with >10 years of experience (n = 2). However, the diagnostic confidence scores showed some variation, particularly for subtle pathologies in the DL4 protocol (Table 4).

#### 3.3.5. Overall Diagnostic Performance

An ROC curve analysis demonstrated excellent results for both protocols. The DL2 protocol achieved an AUC of 0.94 (95% CI: 0.91–0.97), while the DL4 protocol showed an AUC of 0.90 (95% CI: 0.87–0.93). A statistical comparison using DeLong’s test revealed no significant difference between the protocols (*p* = 0.32), suggesting that both acceleration levels maintain clinically acceptable diagnostic accuracy (Figure 2). An overall diagnostic accuracy comparison between the DL2 and DL4 protocols stratified by reader experience, aggregating all pathologies (bone marrow edema, rotator cuff tears, and labral tears), is well documented in Figure 3. Some illustrative clinical cases are shown in Figure 4, Figure 5, Figure 6 and Figure 7, with Figure 7 specifically illustrating a case where pathology detection was compromised in the DL4 protocol compared to the standard and DL2 protocols.

## 4. Discussion

In this study, we focused on demonstrating that DL-driven acquisitions can significantly reduce the acquisition time for shoulder MRI studies without a substantial compromise in diagnostic accuracy. Our findings provide evidence that moderate acceleration (2-fold) maintains diagnostic performance comparable to that of standard protocols, while more aggressive acceleration (4-fold) shows slightly reduced sensitivity for certain pathologies.

The perfect diagnostic performance for bone marrow edema detection across both accelerated protocols aligns with recent findings by Xie et al. [17], who reported comparable sensitivity (98.2%) and specificity (97.9%) between standard and DL-reconstructed sequences. This consistency across studies suggests that DL reconstruction effectively preserves the contrast characteristics necessary for bone marrow pathology evaluation.

Our findings of slightly superior performance with DL2 (AUC = 0.94) compared to DL4 (AUC = 0.90) provide important insights into the optimal acceleration levels for tendon evaluation. These results support Chang and Chow’s [18] emphasis on the “delicate balance between acceleration and fidelity”, particularly for subtle pathologies. This was especially evident in partial-thickness tears, where DL4 showed a minor degradation in sensitivity (96.8% vs. 99.5% for DL2), a finding that parallels Xie et al.’s observation of slightly reduced detection rates for partial-thickness supraspinatus tears in their accelerated protocols.

A novel aspect of this study is the analysis of reader experience’s impact on diagnostic performance, particularly with the DL4 protocol. While Xie et al. reported high inter-reader agreement (κ = 0.82) across their cohort of three readers [17], they did not stratify the results by experience level. Our observation of reduced diagnostic confidence with DL4 among less experienced readers suggests the need for targeted training when implementing higher acceleration protocols. This finding is particularly relevant in labral tear detection, where DL4 showed lower sensitivity (91.7%) compared to DL2 (100%), extending Chang and Chow’s observations regarding the challenges of accelerated protocols in detecting subtle labral pathologies [18].

The maintained high inter-reader agreement across the protocols (W = 0.92–0.98) supports DL reconstruction’s robustness, though the slight reduction in concordance with DL4 suggests that moderate acceleration (2×) might represent an optimal balance between time savings and diagnostic confidence. These findings align with Xie et al.’s conclusion that a “sweet spot” exists in acceleration factors where diagnostic quality is preserved while achieving meaningful time savings [17]. These findings have significant clinical implications, particularly regarding workflow optimization and resource utilization. With the DL2 protocol reducing acquisition times by 50% while maintaining diagnostic accuracy, healthcare centers could potentially increase patient throughput significantly while improving patient comfort and reducing motion artifacts. Chang and Chow specifically highlighted this potential for “democratizing access to MRI” through reduced scan times [18], a vision supported by our findings. To better understand the overall impact of reader experience on diagnostic performance across all evaluated pathologies, we analyzed the aggregated accuracy data for both DL protocols (Figure 3). This analysis revealed that while both protocols maintained high diagnostic accuracy, the performance gap between DL2 and DL4 was more pronounced among less experienced readers, suggesting that expertise may partially compensate for increased acceleration rates.

The consistency in diagnostic accuracy across reader experience levels suggests broad applicability across various clinical settings, from academic centers to community practices. As noted by Xie et al. [17], this robustness is crucial for the widespread implementation of AI-assisted imaging protocols. The potential for reduced scan times to expand MRI accessibility in emergency settings represents another significant advantage, particularly for acute shoulder trauma assessment.

### 4.1. Mechanisms Underlying Reduced Performance with Higher Acceleration

The slight reduction in diagnostic performance observed with the DL4 protocol, particularly for subtle pathologies such as partial-thickness rotator cuff tears and labral lesions, can be attributed to several factors. First, higher acceleration factors naturally result in greater k-space undersampling, reducing the amount of acquired raw data. While DL reconstruction attempts to compensate for this data loss, there are fundamental information theory limits to what can be recovered [12,13,14,15].

Second, the DL reconstruction process at higher acceleration factors may introduce subtle blurring or smoothing effects that can obscure fine structural details that are critical for detecting partial tears. This phenomenon was particularly evident in partial-thickness supraspinatus tears near the footprint, as illustrated in Figure 7. The subtle hyperintensity representing a small articular-sided tear is clearly visible on standard images, slightly less conspicuous on DL2 images, and often not discernible on DL4 images.

Third, the effect of noise amplification at higher acceleration factors, although mitigated by DL reconstruction compared to traditional parallel imaging, may still impact the contrast-to-noise ratio in areas of subtle signal change. This particularly affects structures with inherently lower signals, such as the labrum.

### 4.2. Study Limitations and Future Directions

Several limitations should be considered when interpreting our findings. First, the single-center, single-vendor design substantially limits the generalizability of our results. Different MRI platforms employ distinct DL reconstruction algorithms, which may perform differently at similar acceleration factors. Multi-center, multi-vendor validation studies are essential before widespread clinical implementation.

Second, while methodologically sound, our reference standard based on consensus reading by two experienced radiologists lacks surgical correlation. Although consensus reading by expert radiologists is an established approach for defining imaging reference standards, it cannot account for pathologies that might be missed by standard MRI but detected during surgery. Future studies would benefit from surgical correlation when available, particularly for labral pathologies where MRI has known limitations.

Third, the relatively low prevalence of certain pathologies in our cohort (teres minor tears n = 1, labral lesions n = 9) limits our statistical power for these specific findings. The wide confidence intervals for sensitivity in detecting these lesions reflect this limitation. Larger cohort studies with more balanced pathology distributions or targeted recruitment of specific pathologies would strengthen future investigations.

Fourth, our study focused on a limited set of shoulder pathologies (BME, rotator cuff tears, and labral tears). While these represent common and clinically significant findings, shoulder MRI evaluates numerous other structures and pathologies that warrant investigation.

Fifth, our assessment of reader experience was binary (≤10 years vs. >10 years) and included only four readers in total. A more granular assessment of experience levels with more readers would provide better insights into the learning curve associated with interpreting DL-accelerated images.

Finally, we did not evaluate the long-term clinical impact of implementing DL-accelerated protocols. Studies assessing patient outcomes, workflow efficiency, and cost-effectiveness are needed to fully understand the clinical value of these techniques.

Future research should focus on multi-center, multi-vendor validation studies to establish the generalizability of our findings across different MRI platforms and DL reconstruction algorithms. Also, the evaluation of a broader range of shoulder pathologies, including ligamentous, capsular, and cartilaginous injuries, should be considered, possibly including surgical correlation to better establish the true diagnostic performance of accelerated protocols.

## 5. Conclusions

DL-accelerated shoulder MRI protocols demonstrate high diagnostic accuracy, with DL2 showing performance nearly identical to that of the standard protocol across all evaluated parameters. While DL4 maintains acceptable diagnostic accuracy, it shows a slight degradation in sensitivity for subtle pathologies, particularly among less experienced readers. These findings suggest that the DL2 protocol represents an optimal balance between acquisition time reduction and diagnostic confidence, potentially improving workflow efficiency without compromising diagnostic quality.

Based on our findings, we recommend the following for clinical implementation: (1) The DL2 protocol can be safely implemented for routine shoulder MRI examinations, offering a 50% reduction in acquisition time without compromising diagnostic accuracy. (2) The DL4 protocol should be used with caution, particularly when evaluating patients with suspected subtle pathologies such as partial-thickness rotator cuff tears or labral lesions. (3) When implementing accelerated protocols, institutions should consider providing targeted training for radiologists, especially those with less experience in interpreting DL-reconstructed images.

In conclusion, DL-accelerated shoulder MRI protocols, particularly the DL2 protocol with 2-fold acceleration, offer a promising approach to increase scanner efficiency and patient throughput without compromising diagnostic quality. While more aggressive acceleration with DL4 maintains acceptable performance, the optimal clinical implementation will likely involve tailoring the acceleration factor to the specific clinical question and suspected pathology.

## Figures and Tables

**Figure 1 tomography-11-00048-f001:**
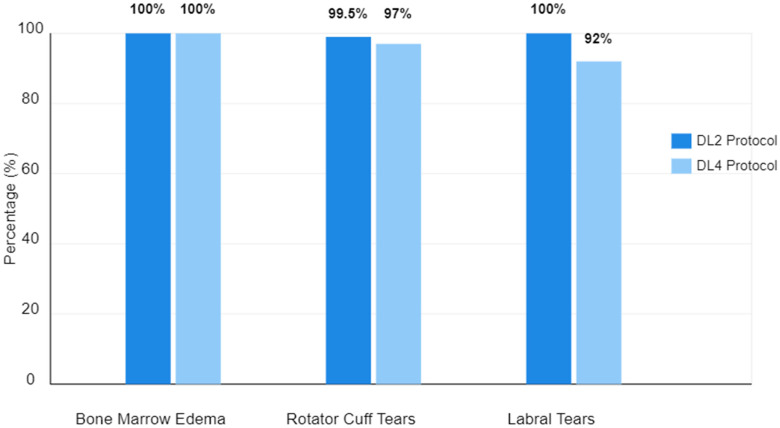
A comparison of the diagnostic performance of the DL2 and DL4 protocols across different pathologies (BME, rotator cuff tears, and labral tears). The bar chart demonstrates the consistently high performance of both protocols, with DL2 showing slightly higher values in some categories.

**Figure 2 tomography-11-00048-f002:**
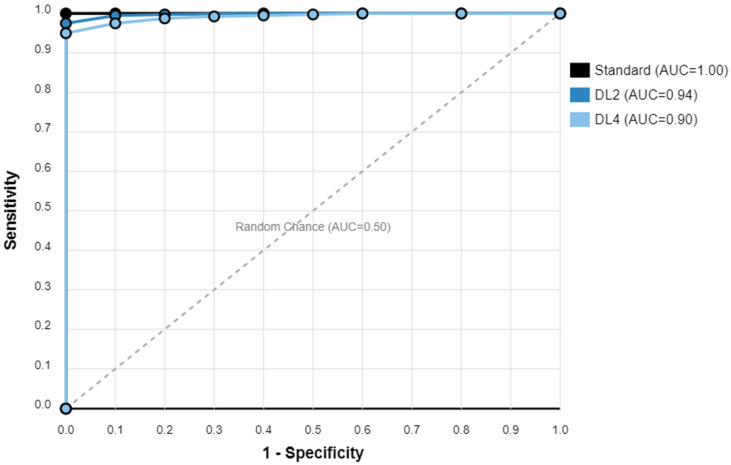
ROC curve comparison with standard reference. ROC (receiver operating characteristic) curves comparing diagnostic performance of standard MRI protocol (black line, AUC = 1.00), DL × 2 protocol (blue line, AUC = 0.94), and DL × 4 protocol (orange line, AUC = 0.90) for rotator cuff tear detection. Dashed diagonal line represents random-chance performance (AUC = 0.50). ROC curves were generated using pooled data from all four readers.

**Figure 3 tomography-11-00048-f003:**
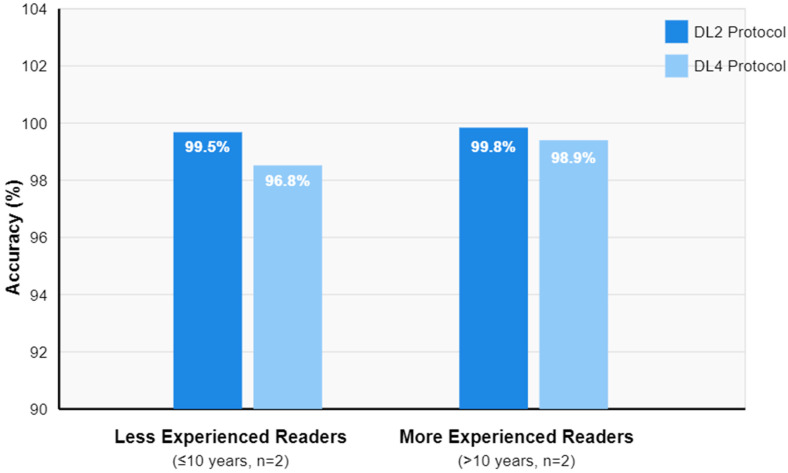
Diagnostic accuracy by reader experience. The graph shows accuracy percentages for less experienced readers (≤10 years, left) and more experienced readers (>10 years, right). The DL × 2 protocol maintained higher overall diagnostic accuracy across both experience levels (≤10 years: 99.5%, >10 years: 99.8%) compared to the DL × 4 protocol (≤10 years: 96.8%, >10 years: 98.9%). Note the smaller accuracy gap between protocols among more experienced readers, suggesting that expertise partially compensates for increased acceleration. Values on the *Y*-axis represent the percentage of diagnostic accuracy.

**Figure 4 tomography-11-00048-f004:**
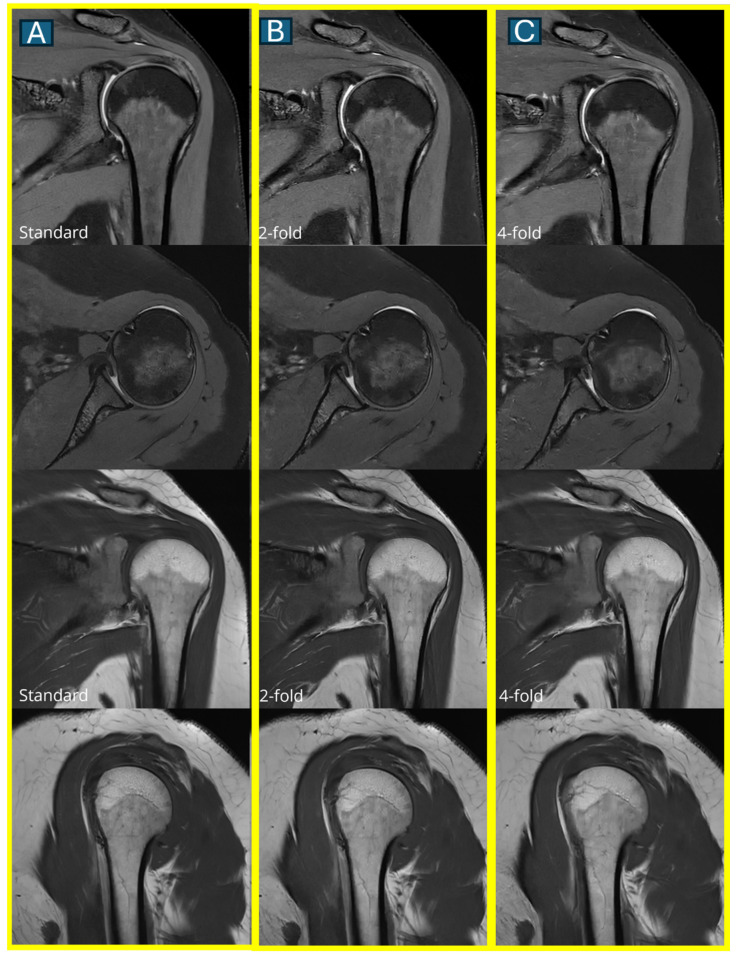
A comparison of image quality across all three protocols. Representative axial and coronal MRI images show a normal rotator cuff in a 45-year-old male. Images in the first column show the standard protocol (**A**), the second column shows the DL2 protocol (2-fold acceleration) (**B**), and the third column shows the DL4 protocol (4-fold acceleration) (**C**). Note the progressive subtle loss of fine detail with increasing acceleration, though all protocols maintain diagnostic quality.

**Figure 5 tomography-11-00048-f005:**
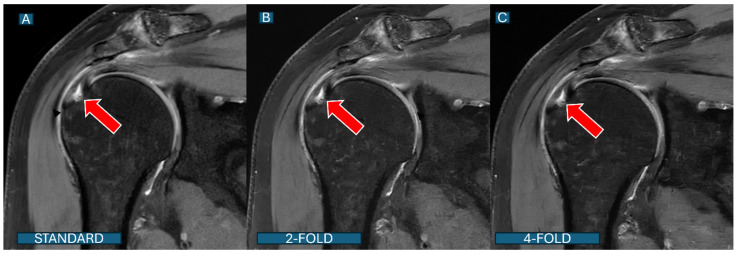
A partial articular-sided supraspinatus tear: a comparison across protocols. Coronal PD fat-saturated images from a 68-year-old patient with shoulder pain demonstrate a partial tear of the supraspinatus tendon (red arrows). The tear is clearly visible with similar diagnostic confidence in all three protocols: (**A**): the standard protocol; (**B**): the DL2 protocol; and (**C**): the DL4 protocol. Note that despite acceleration, both DL protocols maintain excellent visualization of the complete tear morphology.

**Figure 6 tomography-11-00048-f006:**
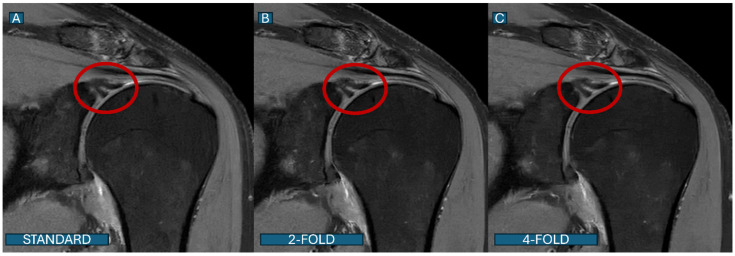
Labral tear detection across protocols. Coronal PD fat-saturated images show a superior labral anterior–posterior (SLAP) lesion (red outline) in a 36-year-old athlete. (**A**): Standard protocol; (**B**): DL2 protocol; (**C**): DL4 protocol. The lesion is well visualized in both the standard and DL2 protocols, while subtle degradation in lesion conspicuity is noted in the DL4 protocol, though the lesion remains clearly detectable.

**Figure 7 tomography-11-00048-f007:**
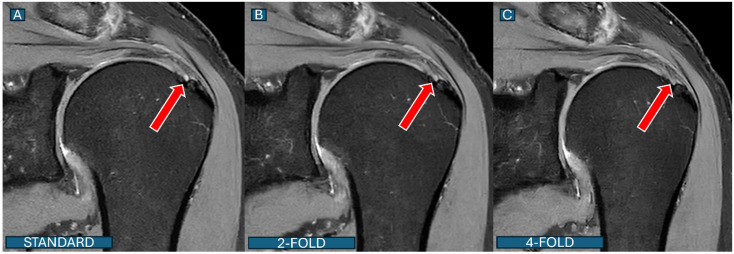
Partial-thickness rotator cuff tear: detection differences across protocols. Coronal fat-saturated PD images of a 52-year-old patient show an articular-sided partial tear of the supraspinatus tendon (red arrows). (**A**): The standard protocol clearly shows the subtle hyperintensity representing a partial tear near the footprint; (**B**): under the DL2 protocol, the tear remains visible but with slightly reduced conspicuity; (**C**): under the DL4 protocol, the subtle tear is substantially less conspicuous and could be missed in clinical interpretation. This case illustrates the potential diagnostic limitations of higher acceleration factors for subtle pathologies.

**Table 1 tomography-11-00048-t001:** Demographic and clinical characteristics of 88 consecutive patients who underwent shoulder MRI with standard and accelerated deep learning protocols. Among the pathological findings, bone marrow edema was identified in 27.3% of cases. The rotator cuff evaluation, performed on a total of 352 tendons (4 tendons per patient), revealed 58 tears (16.5%), with the supraspinatus being the most frequently involved tendon (38/88, 43.2%). Tears were classified as complete or partial for each tendon, showing a balanced distribution in subscapularis involvement (6 complete and 6 partial tears) and a slight predominance of complete tears in the infraspinatus (4 complete vs. 3 partial). The single teres minor tear was complete. Labral pathology was detected in 10.2% of the study population. Values are presented as absolute numbers and percentages unless otherwise specified.

Characteristic	Value
**Study Population**	n = 88
**Gender**	
Male	49 (55.7%)
Female	39 (44.3%)
**Age (years)**	
Mean ± SD	51 ± 14.3
Range	22–78
**Clinical Indication**	
Shoulder pain	88 (100%)
**Pathology Distribution**	
Bone marrow edema	24 (27.3%)
**Rotator Cuff Tears**	58/352 tendons (16.5%)
- **Supraspinatus**	38/88 (43.2%)
Complete tears	18 (20.5%)
Partial tears	20 (22.7%)
- **Subscapularis**	12/88 (13.6%)
Complete tears	6 (6.8%)
Partial tears	6 (6.8%)
- **Infraspinatus**	7/88 (8.0%)
Complete tears	4 (4.5%)
Partial tears	3 (3.4%)
- **Teres Minor**	1/88 (1.1%)
Complete tears	1 (1.1%)
**Labral Tears**	
**Side**	
Right	51 (58.0%)
Left	37 (42.0%)

**Table 2 tomography-11-00048-t002:** Results of the four readers using DL2 and DL4 protocols to diagnose RC tears. Se = sensitivity, Sp = specificity; RC = rotator cuff; R = reader; DL2/4 = 2-/4-fold time reduction; SST = supraspinatus; SSC = subscapularis; IST = infraspinatus; RT = round teres.

Reader	R1	R1	R2	R2	R3	R3	R4	R4
Parameter	DL 2	DL 4	DL 2	DL 4	DL 2	DL 4	DL 2	DL 4
SST	Se 38/38 (100%) Sp 50/50 (100%)	Se 37/38 (97%) Sp 50/50 (100%)	Se 38/38 (100%) Sp 50/50 (100%)	Se 37/38 (97%) Sp 50/50 (100%)	Se 38/38 (100%) Sp 50/50 (100%)	Se 38/38 (100%) Sp 50/50 (100%)	Se 38/38 (100%) Sp 50/50 (100%)	Se 38/38 (100%) Sp 49/50 (98%)
SSC	Se 12/12 (100%) Sp 274/274 (100%)	Se 11/12 (92%) Sp 273/274 (99%)	Se 12/12 (100%) Sp 274/274 (100%)	Se 11/12 (92%) Sp 273/274 (99%)	Se 12/12 (100%) Sp 274/274 (100%)	Se 11/12 (92%) Sp 273/274 (99%)	Se 12/12 (100%) Sp 274/274 (100%)	Se 12/12 (100%) Sp 273/274 (99%)
IST	Se 9/9 (100%) Sp 79/79 (100%)	Se 8/9 (89%) Sp 78/79 (99%)	Se 9/9 (100%) Sp 78/79 (99%)	Se 8/9 (89%) Sp 78/79 (99%)	Se 9/9 (100%) Sp 79/79 (100%)	Se 9/9 (100%) Sp 79/79 (100%)	Se 9/9 (100%) Sp 78/79 (99%)	Se 8/9 (89%) Sp 78/79 (99%)
RT	Se 1/1 (100%) Sp 87/87 (100%)	Se 1/1 (100%) Sp 87/87 (100%)	Se 1/1 (100%) Sp 87/87 (100%)	Se 1/1 (100%) Sp 87/87 (100%)	Se 1/1 (100%) Sp 87/87 (100%)	Se 1/1 (100%) Sp 87/87 (100%)	Se 1/1 (100%) Sp 87/87 (100%)	Se 1/1 (100%) Sp 87/87 (100%)

**Table 3 tomography-11-00048-t003:** Results of the four readers using DL2 and DL4 protocols to diagnose BME, RC tears, and labral tears. Se = sensitivity, Sp = specificity; RC = rotator cuff; R = reader; DL2/4 = 2-/4-fold time reduction; BME = bone marrow edema.

Reader	R1		R2		R3		R4
Parameter	DL 2	DL 4	DL 2	DL 4	DL 2	DL 4	DL 2
BME	Se 24/24 (100%) Sp 64/64 (100%)	Se 24/24 (100%) Sp 64/64 (100%)	Se 24/24 (100%) Sp 64/64 (100%)	Se 24/24 (100%) Sp 64/64 (100%)	Se 24/24 (100%) Sp 64/64 (100%)	Se 24/24 (100%) Sp 64/64 (100%)	Se 24/24 (100%) Sp 64/64 (100%)
RC tears	Se 58/58 (100%) Sp 274/274 (100%)	Se 55/58 (95%) Sp 274/274 (100%)	Se 58/58 (100%) Sp 274/274 (100%)	Se 57/58 (98%) Sp 272/274 (99%)	Se 57/58 (98%) Sp 274/274 (100%)	Se 56/58 (97%) Sp 273/274 (99%)	Se 58/58 (100%) Sp 274/274 (100%)
Labral tears	Se 9/9 (100%) Sp 79/79 (100%)	Se 8/9 (89%) Sp 78/79 (99%)	Se 9/9 (100%) Sp 78/79 (99%)	Se 8/9 (89%) Sp 78/79 (99%)	Se 9/9 (100%) Sp 79/79 (100%)	Se 9/9 (100%) Sp 79/79 (100%)	Se 9/9 (100%) Sp 78/79 (99%)

**Table 4 tomography-11-00048-t004:** Diagnostic confidence score analysis across protocols. Values represent median (IQR). Confidence scale: 4 = definite presence, 3 = probable presence, 2 = possible presence, 1 = definite absence. Readers stratified by experience: experienced (>10 years, n = 2) and less experienced (≤10 years, n = 2). Statistical analysis: Kruskal–Wallis test with Benjamini–Hochberg correction. * *p* < 0.05.

Pathology Type	Standard Protocol	DL2 Protocol	DL4 Protocol	*p*-Value
**Bone Marrow Edema**				
Median (IQR)	3.9 (3.7–4.0)	3.8 (3.5–4.0)	3.7 (3.4–4.0)	0.42
Experienced readers	3.9 (3.8–4.0)	3.8 (3.6–4.0)	3.7 (3.5–4.0)	0.38
Less experienced readers	3.8 (3.7–4.0)	3.7 (3.5–4.0)	3.6 (3.3–3.9)	0.29
**Rotator Cuff Tears**				
Complete tears	3.9 (3.7–4.0)	3.7 (3.4–4.0)	3.6 (3.3–3.9)	0.21
Partial tears	3.8 (3.5–4.0)	3.6 (3.3–3.9)	3.3 (3.0–3.7)	0.038 *
Experienced readers	3.8 (3.6–4.0)	3.7 (3.4–3.9)	3.5 (3.2–3.8)	0.15
Less experienced readers	3.7 (3.4–3.9)	3.5 (3.2–3.8)	3.2 (2.9–3.6)	0.028 *
**Labral Tears**				
Complete tears	3.8 (3.5–4.0)	3.6 (3.3–3.9)	3.4 (3.1–3.7)	0.045 *
Partial tears	3.7 (3.4–3.9)	3.5 (3.2–3.8)	3.3 (3.0–3.6)	0.032 *
Experienced readers	3.8 (3.5–4.0)	3.6 (3.3–3.9)	3.4 (3.1–3.7)	0.040 *
Less experienced readers	3.6 (3.3–3.9)	3.4 (3.1–3.7)	3.2 (2.9–3.5)	0.035 *

## Data Availability

Data is unavailable due to privacy or ethical restrictions.

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
