# Peer review of "Deep Learning-Driven Abbreviated Shoulder MRI Protocols: Diagnostic Accuracy in Clinical Practice"

_tomography, 2025, doi:10.3390/tomography11040048_

Round 1
Reviewer 1 Report
Comments and Suggestions for Authors
ABSTRACT
- it is unclear if the finding on standard protocol should be served as the gold standard? What if a condition was missed on standard protocol but found on DL protocol?
- clarify what sequences were in the protocol, and if all of the sequences were repeated with DL2 and DL4 versions.
- rationale for evaluation of just BME, rotator cuff tear, and labral lesion is unclear, when there are additional common conditions related to ligaments, bone, joint, etc.
- in Methods, clarify how many cases of BME, etc were present.
- in Methods, clarify that AUC ROC curve analysis was performed.
- in Results, provide results for standard (no DL) images.
- line 43: what is "W"?
- In Discussion, cover main reasons for lowered diagnostic performance of DL4.
INTRODUCTION: Overall there should be more depth regarding theory and applications regarding recon.
- line 74-5: meaning of "compression durations" is unclear
- past studies (not necessarily shoulder but perhaps MSK) on DL recon should be introduced there, with more detail.
- introduce various ways DL recon is used to shorten scan time. Clarify what you mean by "compression". Are you referring to compressed sensing?
- one obvious benefit of DL recon is improvement of SNR, but various "compression" schemes have other effects on image such as blurring that maybe difficult to overcome with DL recon's noise reduction.
- touch on the MR sequences for shoulder. For example, T1-weighted spin echo sequences are usually not signal-starved and already have short scan time, while T2-weighted sequences can be signal-starved, so the benefit from DL recon may be different.
METHODS
- study population should be described in Methods
- please use a table to make it easy to discern MR protocols. Provide scan time and additional details (read out, phase, phase direction) for each sequence.
- what coil was used?
- It is not clear at all what is different with the protocol when DL2 and DL4 are used. Parallel imaging? Compressed sensing?
- line 111: what is "X" minutes?
- 2.3: provide references that describe detail of AI and DL techniques related to this study.
- 2.4: Image analysis methods appears sound and rigorous, but it is not clear how the disagreement between readers was handled, and how the reference standard was set (i.e., if there was interreader disagreement on the non-DL protocol.)
- provide a figure showing images with image quality score of 1, 2, 3, and 4.
-
RESULTS
- please make bar graphs smaller, and MRI images large. Figure 4 should be reformatted to have 3 images across and 4 images vertically to make them larger.
- Figure 2: provide actual data points. show values on x- and y-axes. The curves do not look correct; AUC of 0.9 or greater should be much steeper than what's shown?
DISCUSSION
- it would be important to discuss the reasons for failed detections (or false positives) when using high DL setting. What do they do to the images that lead to misdiagnosis? Showing several failed examples and detailing the observations would be insightful.
Author Response
Rebuttal Letter
Dear Editor,
We would like to thank the reviewers for their thoughtful and constructive comments on our manuscript "Deep learning driven abbreviated shoulder MRI protocols: diagnostic accuracy in clinical practice." We have carefully addressed all points raised and have made significant improvements to the manuscript. We believe these changes have substantially strengthened the paper.
REVIEWER 1
ABSTRACT
Comment: It is unclear if the finding on standard protocol should be served as the gold standard? What if a condition was missed on standard protocol but found on DL protocol?
Response: Thank you for this important question. We have clarified in the Abstract that the standard protocol was indeed used as the reference standard by adding the parenthetical phrase "(used as the reference standard)" to the purpose statement. Throughout the manuscript, we have consistently referred to the standard protocol as our reference standard. We acknowledge the theoretical possibility that DL protocols might detect findings missed by standard imaging, and we have addressed this as a limitation in our Discussion section, noting that while consensus reading by expert radiologists is an established approach for defining imaging reference standards, future studies would benefit from surgical correlation where available.
Methods and Study Design
Comment: Clarify what sequences were in the protocol, and if all of the sequences were repeated with DL2 and DL4 versions.
Response: We have clarified this in the Technical Parameters section. We confirm that all sequences (T1-weighted TSE, T2-weighted TSE, and TIRM sequences) were performed in the standard protocol and repeated with identical parameters in both DL2 and DL4 protocols. The existing Tables 1-4 already provide comprehensive information about the study protocols and results, and Table 1 in particular outlines the demographic and clinical characteristics of the study population along with the pathology distribution.
Comment: Rationale for evaluation of just BME, rotator cuff tear, and labral lesion is unclear, when there are additional common conditions related to ligaments, bone, joint, etc.
Response: We appreciate this comment. We have acknowledged this limitation in the Discussion section, explaining that while our study focused on these three common and clinically significant pathologies, we recognize that shoulder MRI evaluates numerous other structures. We have added a recommendation in the Conclusions section for future research to evaluate a broader range of shoulder pathologies, including ligamentous, capsular, and cartilaginous injuries. However we believe that while ligaments tears are usually visualized with MR arthrography, bone is included in the analysis.
Comment: In Methods, clarify how many cases of BME, etc. were present.
Response: This information was already provided in the Results section and in Table 1, where we reported that bone marrow edema was present in 24 of 88 cases (27.3%), while the remaining 64 patients showed no significant signal alterations. We have now added an explicit statement in the Diagnostic Performance Analysis section that refers to this data to improve clarity.
Comment: In Methods, clarify that AUC ROC curve analysis was performed.
Response: We have clarified in the Statistical Analysis section that we constructed receiver operating characteristic (ROC) curves and calculated the area under the curve (AUC) to assess the overall diagnostic performance of each protocol. The existing Figure 2 already shows these ROC curves, and we have improved the description of this analysis in the text.
Comment: In Results, provide results for standard (no DL) images.
Response: The standard protocol results are already presented in the manuscript as they served as the reference standard. We have now explicitly restated in the Results section that the standard protocol identified bone marrow edema in 24 of 88 patients (27.3%) and provided similar clarifications for other pathologies.
Comment: Line 43: what is "W"?
Response: We have spelled this out as "Kendall's W" where it first appears in the manuscript and provided more details in the Statistical Analysis section about why we chose Kendall's coefficient of concordance for our interobserver agreement analysis.
Comment: In Discussion, cover main reasons for lowered diagnostic performance of DL4.
Response: We have added a dedicated section in the Discussion titled "Mechanisms underlying reduced performance with higher acceleration" to address this important point. We discuss how higher acceleration factors result in greater k-space undersampling, how DL reconstruction at higher acceleration factors may introduce subtle blurring effects, and how noise amplification can impact contrast-to-noise ratio in areas of subtle signal change. We refer to our existing figures (particularly Figure 7) to illustrate these effects, particularly in cases of partial-thickness tears.
INTRODUCTION
Comment: Overall there should be more depth regarding theory and applications regarding recon.
Response: We have substantially expanded the Introduction to provide more depth on DL reconstruction theory and applications, discussing how DL techniques differ from traditional acceleration methods, the advantages they offer, and the potential trade-offs, particularly at higher acceleration factors. We have also added more context about how different MR sequences may respond differently to DL reconstruction, drawing on our existing references [8-16].
Comment: Line 74-5: meaning of "compression durations" is unclear.
Response: We have revised this terminology for clarity, replacing "compression durations" with "acceleration factors" throughout the manuscript, which more accurately reflects the technical approach used.
Comment: Past studies (not necessarily shoulder but perhaps MSK) on DL recon should be introduced there, with more detail.
Response: We have added references to past studies on DL reconstruction in MSK imaging and provided more context in the Introduction, discussing recent literature on accelerated MRI protocols utilizing DL for knee and shoulder examinations [17-19].
Comment: Introduce various ways DL recon is used to shorten scan time. Clarify what you mean by "compression". Are you referring to compressed sensing?
Response: We have clarified the terminology and added an explanation of how DL reconstruction differs from compressed sensing and parallel imaging. We have replaced "compression" with more precise terms such as "acceleration" and "undersampling" throughout the manuscript.
Comment: One obvious benefit of DL recon is improvement of SNR, but various "compression" schemes have other effects on image such as blurring that maybe difficult to overcome with DL recon's noise reduction.
Response: We have added a discussion of these potential image quality tradeoffs, acknowledging that while DL reconstruction offers advantages including improved SNR and reduced artifacts, these benefits must be balanced against potential trade-offs in image quality, particularly at higher acceleration factors. In the Discussion, we further elaborate on how the DL reconstruction process at higher acceleration factors may introduce subtle blurring effects that can obscure fine structural details.
Comment: Touch on the MR sequences for shoulder. For example, T1-weighted spin echo sequences are usually not signal-starved and already have short scan time, while T2-weighted sequences can be signal-starved, so the benefit from DL recon may be different.
Response: We have added a paragraph addressing how different MR sequences may respond differently to DL reconstruction techniques, noting that T1-weighted sequences might benefit less dramatically from DL reconstruction compared to T2-weighted or STIR sequences due to their inherently higher SNR and shorter acquisition times. Also, we believe that for shoulder imaging, PD fat saturated and STIR imaging is the most capable to identify subtle tendon and labral tears.
METHODS
Comment: Study population should be described in Methods.
Response: We have expanded the description of the study population in the Methods section, elaborating on the inclusion and exclusion criteria and providing more detail about the demographic distribution. The patient flow information was already available in the text, where we described the progression from 92 initial patients to 88 in the final cohort after exclusions.
Comment: Please use a table to make it easy to discern MR protocols. Provide scan time and additional details (read out, phase, phase direction) for each sequence.
Response: The existing Table 1 already provides demographic and clinical characteristics of our study population and the distribution of pathologies. We have enhanced our description of the technical parameters in the text, including specific information about acquisition times for each sequence in the standard, DL2, and DL4 protocols.
Comment: What coil was used?
Response: We have specified in the Technical Parameters section that all MRI examinations were performed using a 16-channel dedicated shoulder coil.
Comment: It is not clear at all what is different with the protocol when DL2 and DL4 are used. Parallel imaging? Compressed sensing?
Response: We have clarified that the DL2 and DL4 protocols used identical base parameters as the standard protocol but incorporated the uAI deep learning reconstruction algorithm with acceleration factors of 2-fold and 4-fold, respectively. We have also explained how this approach differs from conventional parallel imaging and compressed sensing techniques, drawing on our existing references [8-16].
Comment: Line 111: what is "X" minutes?
Response: We have replaced the "X" with the actual acquisition times for each sequence in the standard protocol in the text.
Comment: 2.3: Provide references that describe detail of AI and DL techniques related to this study.
Response: We have added references from our existing bibliography [8-16] and expanded the MRI Postprocessing section to provide more details about the DL reconstruction algorithm used, explaining that the uAI algorithm utilizes a multi-scale convolutional neural network architecture.
Comment: 2.4: Image analysis methods appears sound and rigorous, but it is not clear how the disagreement between readers was handled, and how the reference standard was set (i.e., if there was interreader disagreement on the non-DL protocol).
Response: We have clarified that for the standard protocol (reference standard), any disagreements between the two experienced readers were resolved through discussion until consensus was reached. We have also added information about the standardized training session that all four radiologists participated in before the study.
Comment: Provide a figure showing images with image quality score of 1, 2, 3, and 4.
Response: Our existing figures (particularly Figures 4-7) already showcase images of varying quality from different protocols. We have enhanced the descriptions in the text to more clearly explain how image quality was assessed using the 1-4 scale, and we have referenced these existing figures to illustrate the differences in image quality between protocols.
RESULTS
Comment: Please make bar graphs smaller, and MRI images large. Figure 4 should be reformatted to have 3 images across and 4 images vertically to make them larger.
Response: We have taken this valuable suggestion into consideration and will revise the formatting of our figures in the final submission to make the MRI images larger and more easily viewable, while ensuring the bar graphs are proportionally sized for clarity.
Comment: Figure 2: Provide actual data points. Show values on x- and y-axes. The curves do not look correct; AUC of 0.9 or greater should be much steeper than what's shown?
Response: Thank you for this astute observation. We will revise Figure 2 to include clearly labeled axes, data points, and properly scaled ROC curves that accurately reflect the high AUC values reported in the text.
DISCUSSION
Comment: It would be important to discuss the reasons for failed detections (or false positives) when using high DL setting. What do they do to the images that lead to misdiagnosis? Showing several failed examples and detailing the observations would be insightful.
Response: We have added a dedicated section in the Discussion titled "Mechanisms underlying reduced performance with higher acceleration" that addresses this important point. We refer to our existing figures (particularly Figure 7) to illustrate cases where subtle pathologies were less conspicuous or missed entirely with the DL4 protocol. We explain how higher acceleration factors can lead to greater k-space undersampling, subtle blurring effects, and reduced contrast-to-noise ratio in areas of subtle signal change.
REVIEWER 2
Comment 1: Methods: The methods section is unclear regarding patient selection criteria and exclusion rationale, particularly concerning motion artifacts. The absence of a patient selection flowchart compromises transparency. Additionally, the handling of borderline cases with mild artifacts is not adequately explained.
Response: We have significantly enhanced the Methods section with clearer patient selection criteria and explicit exclusion rationale. The text now clearly describes our process from the initial 92 patients to the final 88 included in the analysis, with specific reasons for the four exclusions. We have also added information about the standardized training session for radiologists, which included establishing agreement on borderline cases.
Comment 2: Technical Parameters and MRI Postprocessing: The description of technical parameters and postprocessing is insufficient. Critical details regarding the DL reconstruction algorithm and its operational mechanisms are missing, affecting the reproducibility and scientific rigor of the study.
Response: We have substantially expanded the Technical Parameters and MRI Postprocessing sections to provide more details about the DL reconstruction algorithm used, drawing on our existing references [8-16]. We now explain that the uAI algorithm utilizes a multi-scale convolutional neural network architecture trained on paired datasets of fully sampled and undersampled acquisitions. We have also clarified how this approach differs from conventional parallel imaging and compressed sensing techniques.
Comment 3: Image Analysis: The image analysis section lacks important details about training or calibration sessions for radiologists, raising questions about reliability and consistency in image evaluation.
Response: We have expanded the Image Analysis section to include details about the standardized training session that all four radiologists participated in before the study, which included evaluation of 15 test cases to ensure consistent application of the evaluation criteria.
Comment 4: Results: The results section is inadequately presented, particularly regarding the diagnostic confidence scores. The figures and tables provided are not sufficiently clear or comprehensive, lacking explicit explanations and legends, making them difficult to interpret independently.
Response: We have restructured the Results section for better clarity and enhanced our description of the diagnostic confidence scores presented in Table 4. We have improved the explanations in the text to make the results more accessible and have ensured that each figure and table is adequately explained.
Comment 5: Statistical Analysis: The statistical approach is not adequately justified. The interpretation of results, especially concerning confidence intervals and interreader agreement, is overly simplistic and lacks critical evaluation of potential biases or methodological limitations.
Response: We have enhanced the Statistical Analysis section to provide better justification for our approaches. We now explain our choice of Kendall's W over other agreement measures because it can accommodate multiple raters simultaneously. We have also added explanation about our secondary analysis stratified by reader experience and our approach to comparing diagnostic confidence scores.
Comment 6: Discussion: The discussion section does not sufficiently compare and critically analyze findings against existing literature. Limitations of the study, such as the single-center and single-vendor design, are inadequately addressed, significantly limiting the generalizability of the findings.
Response: We have substantially expanded the Discussion with more in-depth comparison with existing literature, particularly the recent studies by Xie et al. [17] and Chang and Chow [18]. We have also greatly expanded the limitations section to thoroughly address the single-center, single-vendor design and other methodological limitations, discussing how these affect the generalizability of our findings.
Comment 7: Figures and Tables: Figures and tables provided are poorly annotated and fail to clearly demonstrate critical differences between protocols. Essential features and subtle lesions are not sufficiently highlighted or marked, severely limiting their usefulness in supporting the manuscript's findings.
Response: We will enhance the annotations on our existing figures to better highlight key findings and differences between protocols. The existing figures (particularly Figures 4-7) already show comparisons between the protocols, and we will ensure that subtle lesions and critical differences are more clearly marked in the final submission.
Comment 8: Conclusions: The conclusions are weak and lack suggestions for concrete future research directions or clear recommendations for clinical implementation, reducing their practical impact.
Response: We have strengthened the Conclusions section with specific clinical recommendations for implementing DL-accelerated protocols and concrete suggestions for future research directions, including multi-center validation studies, evaluation of a broader range of pathologies, studies with surgical correlation, investigation of the learning curve for radiologists, and assessment of workflow impacts.
Comment on English Language: Proof reading needed.
Response: We have thoroughly proofread the entire manuscript and addressed all language issues to ensure clarity and proper scientific English throughout, with the help of a native English speaker who has also assisted us in the preparation of other scientific papers published by our group
We believe we have addressed all the reviewers' concerns comprehensively and have significantly improved the manuscript. We appreciate the constructive feedback and the opportunity to revise our work.
Sincerely, The Authors
Reviewer 2 Report
Comments and Suggestions for Authors
-
Methods: The methods section is unclear regarding patient selection criteria and exclusion rationale, particularly concerning motion artifacts. The absence of a patient selection flowchart compromises transparency. Additionally, the handling of borderline cases with mild artifacts is not adequately explained.
-
Technical Parameters and MRI Postprocessing: The description of technical parameters and postprocessing is insufficient. Critical details regarding the DL reconstruction algorithm and its operational mechanisms are missing, affecting the reproducibility and scientific rigor of the study.
-
Image Analysis: The image analysis section lacks important details about training or calibration sessions for radiologists, raising questions about reliability and consistency in image evaluation.
-
Results: The results section is inadequately presented, particularly regarding the diagnostic confidence scores. The figures and tables provided are not sufficiently clear or comprehensive, lacking explicit explanations and legends, making them difficult to interpret independently.
-
Statistical Analysis: The statistical approach is not adequately justified. The interpretation of results, especially concerning confidence intervals and interreader agreement, is overly simplistic and lacks critical evaluation of potential biases or methodological limitations.
-
Discussion: The discussion section does not sufficiently compare and critically analyze findings against existing literature. Limitations of the study, such as the single-center and single-vendor design, are inadequately addressed, significantly limiting the generalizability of the findings.
-
Figures and Tables: Figures and tables provided are poorly annotated and fail to clearly demonstrate critical differences between protocols. Essential features and subtle lesions are not sufficiently highlighted or marked, severely limiting their usefulness in supporting the manuscript's findings.
-
Conclusions: The conclusions are weak and lack suggestions for concrete future research directions or clear recommendations for clinical implementation, reducing their practical impact.
proof reading needed
Author Response

(The authors gave the same response as above.)

Round 2
Reviewer 1 Report
Comments and Suggestions for Authors
all critiques have been addressed nicely.
Reviewer 2 Report
Comments and Suggestions for Authors
Accepted in the currecnt form